# Spatial Location in Integrated Circuits through Infrared Microscopy [note 1]

**DOI:** 10.3390/s21062175

**Published:** 2021-03-20

**Authors:** Raphaël Abelé, Jean-Luc Damoiseaux, Redouane El Moubtahij, Jean-Marc Boi, Daniele Fronte, Pierre-Yvan Liardet, Djamal Merad

**Affiliations:** 1Laboratoire d’Informatique et Systemes, Aix-Marseille University, 163 Avenue de Luminy, 13288 CEDEX 09 Marseille, France; jean-luc.damoiseaux@lis-lab.fr (J.-L.D.); redouane.el.moubtahij@univ-poitiers.fr (R.E.M.); jean-marc.boi@lis-lab.fr (J.-M.B.); 2STMicroelectronics, 190 Avenue Coq, 13106 Rousset, France; daniele.fronte@st.com; 3eShard, Bâtiment GIENAH, 11 Avenue de Canteranne, 33600 Pessace, France; pierre-yvan.liardet@eshard.com

**Keywords:** thermal imaging, microscopy, secure characterization, pattern location, autofocus, polynomial decomposition, graph matching

## Abstract

In this paper, we present an infrared microscopy based approach for structures’ location in integrated circuits, to automate their secure characterization. The use of an infrared sensor is the key device for internal integrated circuit inspection. Two main issues are addressed. The first concerns the scan of integrated circuits using a motorized optical system composed of an infrared uncooled camera combined with an optical microscope. An automated system is required to focus the conductive tracks under the silicon layer. It is solved by an autofocus system analyzing the infrared images through a discrete polynomial image transform which allows an accurate features detection to build a focus metric robust against specific image degradation inherent to the acquisition context. The second issue concerns the location of structures to be characterized on the conductive tracks. Dealing with a large amount of redundancy and noise, a graph-matching method is presented—discriminating graph labels are developed to overcome the redundancy, while a flexible assignment optimizer solves the inexact matching arising from noises on graphs. The resulting automated location system brings reproducibility for secure characterization of integrated systems, besides accuracy and time speed increase.

## 1. Introduction

Integrated Circuit (IC) (a glossary is present at the end of this paper) are electronic components used in many fields such as wearable technologies and IoT (Internet of Things). The body of these components is made of highly pure silicon, doped with several materials to create a functional network of electronic parts. During the conception process, IC may be inspected using infrared cameras. Silicon has the property of being transparent to infrared light. Specialized cameras embed an indium gallium arsenide (InGaAs) sensor that operates in the Short-Wave Infrared (SWIR) wavelength range from 900 nm to 1700 nm, making it possible to see through the silicon substrates of IC. This kind of camera is widely used in the semiconductor industry as it allows defect detection (among others). In this study, we used such an infrared vision system for the particular purpose of characterizing secure IC through physical hacking attempts.

These IC must be submitted to visual inspections prior to any characterization to identify potential critical parts and locate them. The characterization process then utilizes these locations, taking them as targets for physical hacks. This paper proposes a cartography system to automate spatial positioning on the structures of interest inside an IC, thus overcoming several limitations of the manual way. The proposed system is divided into two complementary sides: an autofocus system to solve the Z-dimension of the cartography and a graph-based location system to solve the XY-dimensions. The overall framework deals with critical imaging constraints implied by the experimental context, such as illumination problems and speckle noise.

### 1.1. Industrial Context

An IC is made of several layers of different materials. In our study, we only consider the two top layers which are: (1) the silicon, (2) a conductive material that forms the conductive tracks and constitutes the heart of the IC (Figure 1). The surface of the conductive material (2) corresponds to the Surface Of Interest (SOI).

Depending on their targeted application, some IC need to be secured to ensure the privacy of users, and their security needs to be validated. The tasks to characterize a secure IC include the study of the IC’s behavior following a physical disruption. Such a disruption can be obtained by targeting a specific location on the IC that corresponds to internal structures such as memories, CPUs or I/O gates, visible on the SOI.

In this study, we addressed the case of local disruptions, which require a high targeting precision.

Any weakness detected during a secure characterization must be addressed by countermeasures and validated by a second characterization. Validation is the key point that raises the question of reproducibility. The conditions of the analysis must be controlled to ensure its reliability, and to exactly reproduce the same conditions for each characterization step. However, mechanical adjustments for the characterization require precise and careful handling, which makes reproducibility difficult. Usually, every security characterization is initiated manually when, on the other hand, the local disruptions must all be conducted with identical setups.

### 1.2. Vision Context

Targeting these disruptions requires nanoscale precision to accurately target structures on the SOI. For that purpose, we used an optical system composed of an infrared camera coupled to an optical microscope to inspect the SOI under high magnification. The camera was SWIR-sensitive, which allowed us to see the metallic materials through the silicon layer. For this vision technology, some third-party equipment was necessary to emit SWIR light, reflected by the scene and caught by the InGaAs sensor. We chose not to use some thermally insulated equipment, even if it improves image quality.

On the one hand, this system gave the necessary precision (±1 μm resolution) to find out the target structures for the secure characterization while keeping the IC functional (non-invasive vision). On the other hand, the result was poor image quality—low-contrasted, gray-leveled and noisy. Speckle noise is inherent to infrared systems; here, it was heavily worsened by optical magnification. Moreover, this not thermally insulated sensor is subject to the thermal inference of the environment, which disturbs the vision process. This thermal inference comes from the third-party equipment that emits the SWIR light, which is reflected by the conductive materials and thus triggers the InGaAs sensor. Additionally, the image quality depends on the material reflectance to light, which may vary with the technology used to build the IC. The random presence of artifacts or contaminants on the silicon layer may also impair the view of the SOI. These contaminants locally block the light casting shadows on the SOI under it (see Figure 2).

Today, the targeting process that precedes characterization is carried out manually. First, the focus of the camera is adjusted once and the target region to be characterized is determined on the SOI.

Depending on the characterization results, these two steps will be repeated to validate countermeasures. Let us highlight two points here:If the IC is tilted or deformed (even by a micron), then the focus may need to be readjusted at every point during the characterization.Re-targeting a structure induces imprecision because the human visual perception can vary significantly.

These points are critical for both accuracy and reproducibility purposes. The use of automated processes to locate structures could solve these issues, but such automated processes must also consider the imaging constraints spotted here.

Self-positioning in a coordinate system implies the overall knowledge of this system. For computer vision, self-positioning can be implemented by scanning the scene to create a coordinate system, whereas locating an object requires visual analysis of this object in the scene. Such a process is split into two sub-processes: 3D scan on one side and pattern recognition on the other side.

In natural scenes, several 3D scanner systems have already been developed, generally using several tools to both take a photo and locate it in space to build an image of the scene [1,2]. The optical system generally used has an infinite depth of field, which enables correctly focused acquisitions that requires no pre-focusing step. A large part of the advances take advantage of merging different specialized acquisition systems to strengthen the accuracy of the overall system [3]. On the shape recognition side, solutions are as numerous as their application fields. Shape recognition is nowadays widely investigated in machine learning approaches [4,5,6,7,8].

However, conventional scanning and pattern-recognition approaches are difficult to apply in the context of IC, as the vision systems lead to great constraints. Several vision systems were used for IC inspection, for example X-ray, thermography, Scanning Electron Microscope (SEM), surface acoustic waves or scanning acoustic microscopy [9,10]. Each system has its own pros and cons in this field, as reviewed in [11], but only non-destructive systems that allow internal inspection are applicable in our study. The closest study [12] proposed to use an SEM to produce high-fidelity images, which permits robust pattern recognition [13] as the obtained images are of far better quality than ours. However, this vision system is incompatible with the characterization field as it is destructive. As far as we know and according to our constraints, no image-based method has been proposed to reliably scan the internals of an IC and locate complex structures.

The aim of this paper is to propose a method for overcoming the image quality constraints and automating the XYZ scanning system to address the issue of reproducibility in IC characterization (cf. Section 1.1). In that context, the use of a single uncooled InfraRed (IR) camera with optical magnifiers and without any other third-party material makes our system innovative and solves a great issue.

For that purpose, we firstly present a specialized autofocus system that can efficiently scan the conductive tracks of an IC (Section 2). Secondly, we discuss a graph-matching-based method to automatically locate the structures of interest to be characterized (Section 3). Section 4 presents the resulting framework, and shows that the system we built is relevant and that it is compatible with infrared microscopy for IC.

## 2. Scanning System for Viewing Integrated Circuits

For characterization, we placed an IC in a bench as illustrated in Figure 3. Let XYZ be the coordinate system where XY correspond to the orthogonal axes of the motors in the horizontal plane, and *Z* is the distance from the optical system to the XY plane. The optical system orientation is fixed vertically along *Z*.

In the context of infrared microscopy, the scan process consists of an iterative XYZ stepping procedure where each step follows an exposition time of the optical system to acquire the best possible image. After the iterative stepping procedure, the images are brought together to create a global image of the scanned object. As the optical system is not precise enough to distinguish the height differences between the internal IC structures, the SOI is considered as a flat surface. During manual prepositioning, the IC happens to be tilted due to an imperfect adjustment. Moreover, because of conception constraints, the SOI can appear as curved. Figure 4 shows examples of possible topologies.

As in any microscopy system, the more magnified the light, the shorter the focal distance and the shorter the field depth, hence the more important the precision of the focus. Therefore, the focus adjustment is of main interest for an accurate scan under high magnification conditions. In addition, this focus adjustment has to be computed on the fly during the XY scan process. Hence, we propose to compute the *Z* focus adjustment with an AutoFocus (AF) system.

### 2.1. Autofocus Methods

The AF mechanism is a deterministic algorithm used to compute the lens position such that the system is focused on the scene/object of interest. If focused, the image of the scene is sharpest. Two types of AF approach exist—active and passive [14]. An active approach depends on an additional system measuring the distance from the lens to the scene. Knowing the optics-to-object distance and the focal length of the system, the correct lens position is estimated to obtain a focused image [15,16,17,18]. As we do not have such a system, we consider passive approaches, which rely on analyzing the images from the optical system to determine the best distance to the scene. Images should be evaluated as the Human Visual System (HVS) does. Then, the main difficulty is the interpretation of human subjectivity through algorithms that follow objective rules and criteria [19].

Over the years, several AF methods have been proposed in different domains, each one adapted to its specific context such as images of natural scenes, low-contrast images, microscopy, digital holography and so on [20,21,22]. In each case the concept of sharpness has to be defined to build an objective function. This function, or focus metric, can rely on statistics on the image data in the spatial, frequency and time-frequency domains. Under natural light microscopy, [23] proposed a method to measure the blurriness of calibration charts based on Bayes-Spectral-Entropy of Discrete Cosine transformed images. The Discrete Cosine Transform was also used in [24,25] to measure the sharpness level, distinguishing high frequencies from medium and low frequencies. A time-frequency analysis was proposed in [26] through the Wavelet Packet Transform, used for its ability to rise high frequencies which characterize the sharpness in their context. A comparison between sixteen AF methods for microscopy was made in [27], with histogram, intensity, statistic, derivative and transform-based methods. Each of these methods were proposed to match specific contexts and overcome the inherent constraints. These constraints do not match ours, for which a first approach was propose in [28] based on wavelet decomposition. However, this approach shows weaknesses when the acquisition context gets worse as it leads to noisier graphs. Image pre-processing could be a solution to this issue [29], but it is time-consuming and, therefore, not a preferred solution in our industrial context. We will see in the next section, how we propose to circumvent this issue of noisy graphs.

### 2.2. Specialized Autofocus for Viewing Integrated Circuits

In our context of focusing on an IC, infrared microscopy raises several constraints. The optical system produces images with low contrast where noise increases with magnification, and where random black spots are visible on the SOI (cf. Section 1.2). These two points make image analysis difficult and make usual AF methods ineffective.

To deal with these specific constraints, an AF was proposed in [30] based on special IC viewing features, analyzed in the time-frequency domain. The Focus Metric based on POlynomial Decomposition (FMPOD) is proposed to evaluate the focus level of images acquired for a varying distance of the optical system from the SOI.

More than conventional multi-resolution analysis such as wavelet transforms, the polynomial transform is a flexible tool used to extract oriented features for a given frequency and scale. Such a transform consists in projecting an image onto a complete basis of orthogonal polynomials. A polynomial basis is defined on the Hilbert vector space equipped with a scalar product such that:(1)$$\langle f_{1}|f_{2}\rangle =\overset{}{\underset{\Omega }{\;\int \int \;}}f_{1}\left(x\right)f_{2}\left(x\right)\omega \left(x\right)dx,$$
where $\langle f_{1}|f_{2}\rangle$ represents the scalar product of functions $f_{1}$ and $f_{2}$, and ω is a weighting function with ∀x∈Ω,ω(x)≥0. The basis is composed of bivariate polynomials:(2)$$P_{d_{1},d_{2}}\left(x\right)=\sum_{\begin{array}{c} \left(d_{1},d_{2}\right)\in {\left[0;d\right]}^{2}\\ d_{1}+d_{2}\leq d \end{array}}a_{d_{1},d_{2}}x_{1}^{d_{1}}x_{2}^{d_{2}},$$
where $d_{1}$ and $d_{2}$ are the degrees of variables $x_{1}$ and $x_{2}$, respectively, and $a_{d_{1},d_{2}}$ is the group of real coefficients of the polynomial. To ensure the orthogonality of the basis, its polynomials meet the scalar product condition:(3)$${\langle P_{d_{1},d_{2}}|P_{l_{1},l_{2}}\rangle }_{\omega }=\left\{\begin{array}{c} 0\;\;\;\;\;\;\;\;\;\;\mathrm{if}\left(d_{1},d_{2}\right)\neq \left(l_{1},l_{2}\right)\\ 1\;\;\;\;\;\;\;\;\;\;\mathrm{if}\left(d_{1},d_{2}\right)=\left(l_{1},l_{2}\right). \end{array}\right.$$

This weighting function defines the basis type (e.g., Legendre, Hermite or Laguerre). Such a basis is noted $B^{D,\omega }$, where its degree $D=sup(d_{1},d_{2})$ and such that:(4)$$B^{D,\omega }={\left\{P_{d_{1},d_{2}}\right\}}_{d_{1},d_{2}\leq D\in N}.$$

As the projection of an image *I* on $B^{D,\omega }$ is modeled by the scalar product of this image by the base, it is essential to determine ω for the decomposition expectation and analysis.

Introduced in [31], polynomial image decomposition was used for its ability to extract textural features from images [32,33,34,35,36,37]. This decomposition is highly customizable by the choice of the basis type, its degree, and the discrete domain on which the projection is calculated.

Indeed, the whole polynomial transform process is discretized according to the multi-resolution analysis scheme [38]. The discrete process to transform a 2D function *U* defined on a domain Ω of size $n_{1}\times n_{2}$ is as follows:Given a resolution factor *L*, we define sub-domains of $\Omega^{L}$ such that:
(5)$$\Omega_{h_{1},h_{2}}^{L}={\left\{x\left(i,j\right)=(x_{1}\left(i,j\right),x_{2}\left(i,j\right))\right\}}_{\left(i,j\right)\in D_{h_{1},h_{2}}^{L}},$$
with $\left(h_{1},h_{2}\right)$ the sub-domain indices, whose size is:
(6)$$D_{h_{1},h_{2}}^{L}=\left[h_{1}\frac{n_{1}}{2^{L}},\left(h_{1}+1\right)\frac{n_{1}}{2^{L}}-1\right]\times \left[h_{2}\frac{n_{2}}{2^{L}},\left(h_{2}+1\right)\frac{n_{2}}{2^{L}}-1\right]$$
and such that $U_{h_{1},h_{2}}^{L}\Omega_{h_{1},h_{2}}^{L}=\Omega$.For each $\Omega_{h_{1},h_{2}}^{L}$, the projection coefficients are given by:
(7)$$b_{d_{1},d_{2}}^{L,h_{1},h_{2}}=\langle U^{L,h_{1},h_{2}}|B_{d_{1},d_{2}}^{L},\rangle$$
where $U^{L,h_{1},h_{2}}$ is the restriction of *U* in the sub-domain $\Omega_{h_{1},h_{2}}^{L}$ and $B_{d_{1},d_{2}}^{L}$ the polynomials defined in given sub-domain,The resulting multi-resolution structure is designed by grouping the projection coefficients according to polynomial degree from the basis used for the projection.

In [30], the use of a Legendre polynomial for the FMPOD is justified by its ability to raise features of interest in infrared images of IC. The authors propose a focus metric by analyzing the directional features highlighted through polynomial decomposition, features considered as relevant to describe a focused image. They found that specific polynomial combinations used from the basis allow the analysis of different directional features. The proposed focus metric is computed from the standard deviation of horizontal and vertical coefficients of this decomposition. It gives robustness to the focus algorithm regardless of the scale, contrasts and lightning, even despite several image degradations.

Figure 5 summarizes the autofocus process using FMPOD.

The FMPOD is robust against impulsive noise and other image degradations that typically occur for infrared IC viewing. Moreover, it is scale-invariant since it stays accurate whatever the magnification. Thus, this autofocus method allows the SOI to be accurately scanned to produce an overall view of it in the IR field. This view enables the automation of the targeting process conducted to disturb an IC structure during a characterization campaign. For that purpose, we investigated a pattern-recognition method to automate the location and the self-positioning of the targeting system on a given structure inside the IC.

## 3. Pattern Recognition

Today, pattern recognition is widely used with the development of learning-based methods, whose performance is well established. However, such approaches cannot apply, as no databases exist about IC structures in our context of infrared viewing. This issue is mostly due to the trade secret policies, but also to the fact that, in characterization processes, the studied IC may be prototypes designed with new technologies. On the opposite, methods based on signal analysis (e.g., cross-correlation [39]) are efficient for optimal recognition contexts where a template image exactly occurs in a target image, even though linear transforms can be tolerated (such as scale or rotation). In our study, a structure to locate in a target IC is subjected to linear transforms like scale, rotation and intensity variation: from one IC to the next, the material properties may imply less contrasts compared to the reference IC. Moreover, contaminants described in Section 1.2 could randomly and partially hide structure features (cf. Figure 2). To overcome all these constraints, we investigated another matching method based on graph theory. We studied graphs for pattern recognition in digital imaging, as template retrieval can be translated in terms of graph matching. Interpreting images as graphs allowed us to give a high abstraction level to problems and to break free from linear transforms applied to images. It then allowed us to overcome the context of image acquisition that may vary substantially. Above all, the key advantage of using graphs comes from the abstraction level of this method—identifying a pattern from a simple scheme of an image, which overrides the need for its imaging view. Indeed, in graph matching, we no longer consider images as we compare graphs, whatever the data they represent.

### 3.1. Graph-Based Methods

For decades, graphs have been studied for pattern recognition in digital imaging, as template retrieval in an image can be translated in term of graph matching. The first and basic approach for graph matching was to consider exact graph matching, that is, the one-to-one correspondence between the nodes of one graph and those of a similar graph modified according to a linear function. Practically, for image analysis, outliers could occur as a consequence of any kind of deformation. Outlier stands for unexpected node or edge, being modified, added or deleted. This case is dealt with inexact Graph Matching (GM) methods, where matching is approximated. Therefore, the problem is redefined as finding the best match between two sets of nodes and edges.

Several studies translate graphs into string- or path-like representations using methods such as random walks [40,41,42] or eigenvector decomposition of a graph adjacency matrix [43]. Strongly inspired from the Levenstein string-edit-distance [44], the Graph Edit Distance (GED) between two paths is defined by the structural modification needed to obtain one from the other (deleting, adding or modifying nodes and/or edges). Finding the path or sub-path that minimizes the edit cost is equivalent to finding the best matching nodes between graphs. Recently, the GED was reformulated into a Quadratic Assignement Problem (QAP), which is usually used for GM methods [45].

QAP was designed to solve the assignment of items from two different groups, while minimizing the global cost of each assignment. A general formulation of QAP, still used nowadays for graph matching, was proposed by Lawler [46] and allows the matching problem to be formulated as the maximization of an objective function:(8)$$J_{g}m\left(X\right)=X^{T}*K*X,$$
where *X* is a binary correspondence matrix and *K* is a second-order affinity matrix encoding nodes and edges as pairwise similarities. With the aim of achieving better robustness against deformations, several methods were proposed, which consider features of higher orders to build the objective function, and transpose the graph topology into constraints for matching optimization. The problem is that the objective function is not convex, which makes an optimal global solution hard to achieve. Several methods proposed to relax the problem on the convex space through the Semi-Definite Programming (SDP) method, which ensures an optimal solution [47]. Despite its mathematical consistency, this approach suffers from its high complexity, critical for large graphs. On the opposite, methods like graduated assignment [46,48,49] try to approximate the solution using an ascent gradient. The approximation is guided by a graduated non-convexity procedure that aims to avoid sticking to local optima. Another approach proposed in [50], the path-following strategy, alternates convex and concave relaxations to gradually reach an optimal solution. Solutions reached by path-following methods are competitive in terms of precision compared to graduated assignments and attempts at simplifying their complexity are still ongoing in the doubly-stochastic domain [51].

Other graph-matching methods avoid the quadratic form of the assignment problem by elevating the graph formulation to a higher subjectivity level. For example, this is the case of methods based on hypergraphs such as described in [52,53], formulated by similarity tensors to create high-order features. In this way, the assignment is solved with linear approximation functions. A recent method [54] was proposed to compute contextual similarities between pairwise nodes and edges using a remastered random walk. This gives high-order features without dealing with hypergraphs, but still linearly solves the assignment. These higher-order methods avoid the quadratic optimization problem, which is an interesting point. For a larger review of the assignment optimization, the reader can refer to [55,56,57].

### 3.2. Application to Integrated Circuits

In the case of infrared microscopy applied to IC, the recognition of patterns faces two main difficulties. The first is inherent to the data topology, while the second comes from the imaging system. On the one hand, to answer conception constraints such as space and simplicity, the structures visible on the SOI correspond to rectangular shapes. As these structures can be basically described as orthogonal connected edges, data are highly redundant. From this description, it is easy to build graphs, but discriminating features have to be defined. We propose a solution in Section 3.2.1. On the other hand, our infrared images are poorly contrasted, noisy, and can be degraded by contaminants. Consequently, interpreting these images can lead to wrong graphs with outliers. Graphs containing outliers can be matched by using an inexact graph matching algorithm, which must also be compliant with the sub-graph matching technique to efficiently locate a small template graph in a larger one (approximately 10 nodes against 1000).

Section 3.2.2 describes such a flexible method.

#### 3.2.1. From Integrated-Circuit Image to Labeled Graph

Let $G^{T}=\left(N^{T},E^{T},L^{T}\right)$ and $G^{IC}=\left(N^{I},E^{I},L^{I}\right)$ be two labeled graphs (directed or not), where *N* represents a set of nodes, *E* a set of edges, and *L* a set of labels for nodes and edges. $G^{T}$ and $G^{IC}$ correspond to the template image and the IC image, respectively.

Considering electronic structures as characterized by horizontal and vertical connected components, we pre-process our images by:anisotropic-like filtering to reduce the infrared granular noise, following the method proposed in [34], based on polynomial decomposition and of an image and its adaptive reconstruction,binarizing using an adaptive Gaussian thresholding,skeletonizing based on the distance transform [58] and its ridge extraction.

We decompose the thus obtained skeleton into a sum of linear components using the Hough transform [59], and we create nodes as the intersections between two components. Edges are created to link nodes if they belong to the same component. A graph can therefore be a disconnected graph containing connected subgraphs if some electronic structures are fully disconnected from the others. As theses graphs are topologically highly redundant due to the design of IC, labels must be computed to discriminate nodes and edges. In the following section we describe two textural and structural descriptors.

##### Structural Descriptor

Through this descriptor, we aim to synthesize the structural features of a local connexity relative to nodes and edges. For that purpose, let *S* be a superincreasing series of p∈N
*s.t.*
(9)$$p_{k+1}\in S\Leftrightarrow p_{k+1}>\sum_{i=0}^{k}p_{k}.$$

For each node and edge in the graph, the proposed descriptor consists of a single value computed as a weighted sum of edge lengths connected to an element. It is important to order the edges by length. In other words,

Considering a node v∈N and $E^{v}\subset E$, the set of its *x* connected edges, ordered by length its structural descriptor is given by
(10)$$\alpha_{v}^{S}=\sum_{i=0}^{x}\left|\right|E_{i}^{v}\left|\right|*p_{i},$$
where $\left|\right|E_{i}^{v}\left|\right|$ is the length of edge $E_{i}^{v}$.Considering an edge e∈E and $E^{v}\subset E$, the set of its *y* connected edges, ordered by length its structural descriptor is given by
(11)$$\beta_{e}^{S}=\sum_{i=0}^{y}\left|\right|E_{i}^{e}\left|\right|*p_{i}.$$

Finally, for scale-adaptive purpose, each element of the sum is normalized with respect to the greatest one. This process is represented for both nodes and edges in Table 1.

Supercrescent series were originally introduced in 1978 in the context of data encryption keys. They led to the creation of the first existing cryptographic systems, enjoyed for their simplicity, but quickly outperformed by advances in cryptography. In our context, the use of such a series is original and allows features to be stored to a single value. Indeed, thanks to the ordering, the weighting allows each value to be saved at a specific scale. Finally, this kind of descriptor takes the step over classical containers such as vectors due to its simplicity and storage requirement. Moreover, as we will see in Section 3.2.2, it is far easier to compare values than vectors.

##### Textural Descriptor

The Histogram of Oriented Gradients (HOG) constitutes a powerful tool for texture description and it has been used extensively over the years. Its effectiveness has been proven by its many applications in shape recognition [60,61,62,62] thanks to its robust recognition abilities and its ease of use. The textural descriptor used in our study concerns an histogram of gradients containing nine bins covering the $\left[0,180\right]$ degree interval, according to [60]. This descriptor is computed for a window around each node, as they correspond to joint points between electronic line segments. In an invariance purpose,

each window is centered on its corresponding node and of a size related to its smallest connected edges,if *d* is the main orientation provided by its connected edges, the gradients in the window are oriented according to *d*,the window is split into four sub-windows and each of the gradient intensity is weighted according to the global window intensity so that each of their histogram is less sensitive to non-linear brightness [60]. Number four is related to the maximal node connexity (at most four neighbors).

The resulting descriptor is then a vector of 4×9 bins. The computation rules are represented in Figure 6.

Finally, these two descriptors are used as labels to add discriminative data for nodes and edges. In that way,

for each node v∈N, its label $\alpha_{v}$ is bipartite such that $\alpha_{v}=\left[\alpha_{v}^{S},\alpha_{v}^{H}\right]$for each edge e∈E, its label $\beta_{e}$ is such that $\beta_{e}=\beta_{v}^{S}$.

By this labeling, we aim to collect as much available and relevant information as possible to complete our graphs. Since we are now able to transpose electronic structures as graphs, we can locate any template through graph matching methods.

#### 3.2.2. Matching Method for Integrated Pattern Location

In our study, we need to find some structures specific to IC in IR images. Let us summarize the constraints we have to deal with: graphs can be very noisy; they are structurally very redundant and a the large size difference between graphs implies a huge number of outliers. Let $G^{T}$ correspond to an IR image containing a partial or full structure of an IC image, and let $G^{IC}$ be the IR image of an IC (potentially unknown) to be characterized. The latter potentially contains a noisy version of the template. $G^{T}$ is likely to be much smaller than $G^{IC}$ (in size and order).

Among the numerous graph matching methods proposed over the years, we found that the method proposed by Dutta et al. [54] matches our constraints. Indeed, their approach is based on the computation of higher order similarities between pairwise nodes and edges of input graphs. To that end, they computed a Tensor Product Graph (PG), whose nodes and edges consistently model potential matching across the two graphs. They computed a similarity measure for each element of this graph, and applied a diffusion process to it. Then, they computed new contextual similarities from this diffusion process. This approach takes the advantages of the hypergraphs-based techniques but without the complexity due to the use of costly manipulation of tensors. Thanks to this formulation, the matching problem is reduced to a linear optimization to which constraints may be added as the number of matches per node, the incidence degree of two associated nodes and so on [63]. We propose investigating this approach for our context, together with the discriminating features previously described. We also propose a numerical threshold to finally validate a match.

The PG of two graphs $G^{p}$ and $G^{q}$ is represented by the ⊗ operator, such that $G^{pq}=G^{p}⊗G^{q}$, with $G^{pq}=\left\{V^{pq},E^{pq}\right\}$, where $V^{pq}=V^{p}\times V^{q}$ and $E^{pq}=\left\{\left(\left(v_{i}^{p},v_{i}^{q}\right),\left(v_{j}^{p},v_{j}^{q}\right)\right)|\left(v_{i}^{p},v_{j}^{p}\right)\in E^{p},\left(v_{i}^{q},v_{j}^{q}\right)\in E^{q}\right\}$. Each resulting node and edge correspond to a structurally possible pairwise match in the assignment problem (as an association graph does). Strong contextual information over nodes and edges can be computed from a PG, using a diffusion method [64] or a random walk [65] to produce neighborhood context information. In the second case and in a similar way as in [66], once pairwise similarities are used as transition probabilities of a walk, the probabilities of a walker ending on node *n* at instant *t* are used to compute a new contextual similarity. In [54], the backtrackless random walk is computed on the PG, from which the new contextual similarity is strengthened compared to usual walks. It is proven that the backtrackless walk has a more discriminating power of the resulting contextual similarities [67].

In our study, according to the labeling system previously proposed, similarities $S_{n}$ between nodes $n^{T}$, $n^{IC}$ and $S_{e}$ between edges $e^{T}$, $e^{IC}$ are given by:(12)$$S_{n}\left(n_{1},n_{2}\right)=\frac{\exp^{-|L_{n_{1}}^{1}-L_{n_{2}}^{1}|}+\exp^{-|corr(L_{n_{1}}^{2},L_{n_{2}}^{2})|}}{2}$$
(13)$$S_{e}\left(e_{1},e_{2}\right)=\exp^{-|L_{e_{1}}-L_{e_{2}}|}.$$

To prevent confusion, note that *e* is used for edges and not for exponents (exp). Finally, these pairwise similarities are reevaluated by the backtrackless walk computed on the association graph $G_{A}=G^{T}⊗G^{IC}$. Acting like a diffusion process, $S_{n}$ and $S_{e}$ now incorporate contextual information. Once computed, the assignment problem can be reduced to a node and edge selection from $G_{A}$ according to the contextual similarity and is solved as a constrained linear optimization in polynomial time. Finally, the selected nodes and edges from $G_{A}$, namely $N_{sol}$ and $E_{sol}$, constitute the best match.

However, the selected nodes and edges could form disconnected sub-graphs, as:input graphs may be disconnected and PG preserves connectivity,no optimization condition constrains any global connectivity in the solution.

Then, a template graph could be matched to several sub-graphs in the target graph. This flexibility is necessary in noisy graphs. To find the final template location, we consider the largest connected sub-graph among every matched sub-graph. The template graph location is evaluated by projecting it on the final matched sub-graph. This selection is submitted to a numerical validation by computing the similarity rate:(14)$$S_{tot}=\frac{\sum_{i=0}^{x}S_{n}\left(n_{i}^{A}\right)}{2x}+\frac{\sum_{j=0}^{y}S_{e}\left(e_{j}^{A}\right)}{2y},$$
where *x* is the size of the node set $\left\{n^{A}\right\}\subset N_{sol}$ and *y* the size of the edge set $\left\{e^{A}\right\}\subset E_{sol}$. These two sets form the selected sub-graph. If $S_{tot}$ is greater than a critical threshold, this solution is validated.

In order to test the robustness of such a location process (i.e., graph matching and projection), we present experimental results over a large data set in the next section.

#### 3.2.3. Location Method Validation

As far as we know, there exists no database of infrared views of integrated circuits. Moreover, the trade secret context does not allow us to create one. For that reason, we validated our method on synthetic data. In [68], a data set is built from synthetic images that match the practical constraints encountered in our context, where noisy graphs occur. In line with the design of IC, electronic structures are simulated through combination of random but orthogonal white lines. These templates are noised with a random multiplicative noise to simulate thermal noise. Synthetic IC are built from the combination of several templates that are scaled and rotated in black images. The resulting database is then composed of synthetic IC, on which we test our location framework of synthetic templates.

In this experiment, successful location corresponds to both a correct graph matching and a correct projection of this graph on the target one. In [68], we compared the success rate of our framework with the method proposed in [54], based on PG. Their overall performances are presented as true/false rates in Figure 7, where the area under each curve represents the overall location success rate for the corresponding method. The detection rate of the templates following our method is evaluated at approximately 97%, outperforming the one proposed by Dutta et al.

We did not make a comparison with additional methods, as this is not the purpose of this paper. The interested reader can refer to [51,53,54] for a benchmark of several methods. Also, if some other methods could be relevant for our experiment, few of them match our overall constraints of flexibility to accept outliers, be robust against noise and solve the problem in polynomial time.

From this validation, we present some practical experiments in Section 4. More specifically, the experiments include template location for two particular cases allowed by our graph-based approach—(1) both template and target images are real images; (2) the target image is a real image, and the template image is synthetic. This second case is critical, as only workable through graph matching approaches. Moreover, it makes sense in the experimental context that prototypes may include unknown structures, for which no image acquisition was ever made. Locating an electronic structure from its theoretical shape is then necessary for prototype characterization.

## 4. Experiments & Results

Implemented for IC secure characterization, the whole process presented in this paper brings automation and reproducibility where human driven tasks are usually required. In this section, we describe the experiment from the scan of an IC to characterize to the location of the structures of interest.

In this case of study, we positioned an IC under the optical system such that the SOI was visible through the silicon layer. To optimize the runtime of the scan process, we first precomputed a focus map using the autofocus system presented in Section 2. Several autofocuses were processed on the SOI. From these samples, we approximated a focus map by projection onto a 3D Gaussian surface, which is the surface that better fits the topology of such an SOI compared to a simple quadratic surface. This approximation step allowed us to refine the autofocus while keeping an overall reasonable runtime (around 2 min). Furthermore, this limited the impact of the lack of precision of the autofocus due to the presence of contaminants and/or the absence of details in the field of view to help determine whether the focus is correct or not. Figure 8 shows an example of focus map.

With this focus map, the image acquisition was carried out with a 0.5 s exposure time of the optical system in the iterative stepping scan to cover the entire SOI. To prevent possible lack of precision induced by motor stepping, the assembly process cross correlated the overlapping image borders with those of the 4 surrounding images (neighbors). The final image was obtained when each image was stitched to its neighbors with the best alignment. Figure 9 presents the resulting scan of the SOI in the infrared field.

We built our graphs from the processed IR image of IC. At first, we applied an anisotropic smoothing through the polynomial decomposition method proposed in [34]. Then, we binarized the image using a local adaptive thresholding, and we detected the horizontal and the vertical components of the binary image. The graph $G^{IC}$ was built from the connectivity analysis of these horizontal and vertical components. Similarly, the graph $G^{T}$ was built from the IR image of a structure to detect in the IC. The graph-matching based pattern location presented in Section 3 was used to detect $G^{T}$ in $G^{IC}$ (if the corresponding structure was present). In the followings we show the location result for two use cases where the template graph was created from:(1)a photo acquisition of an electronic structure,(2)a synthetic image representing an electronic structure.

The first is a classical case where the template is known and has already been encountered, whereas the second case occurs if the template is only theoretically known (for example a prototype). In both cases, the target graph was created from a real IC view. However, for the sake of clarity, the presented images show partial IC views, as the corresponding graphs are dense and contain many details.

We experimented case (1) for the input template and IC illustrated in Figure 10. The location of this template is presented in Figure 11, where the template projection is confirmed by the similarity rate $S_{tot}=70\%$.

To validate the relevance of our location method in our context, we experimented case (2) where the provided template was a scheme. Figure 12 illustrates the input images while Figure 13 presents the location result, which was numerically validated by $S_{tot}=61\%$. The small number of highlighted matched nodes was a consequence of outliers between the schematic template and the target. This revealed a risk of lack of connectivity in the matching resolution.

Both experiments were conducted successfully. Validation over a synthetic database and the experiments presented here may constitute sufficient evidence of the validity of our method and good reason to continue investigating it: the presented results, from the IC viewing system to the template location system, show that our framework achieved the purpose of this study.

## 5. Conclusions

In the context of IC secure characterization, several tasks “traditionally” accomplished with the human visual system can be automated. This paper proposes state-of-the-art methods, compatible with infrared microscopy constraints, for automating structure recognition (location) in an integrated circuit for the secure characterization process. To that aim, we proposed a framework that uses different tools to scan conductive tracks in an integrated circuit and locate the patterns to be characterized. To conduct an infrared microscopy scan of the integrated circuit we used a specialized autofocus system that analyzes image features through a discrete polynomial image transformation. Locating the structures of interest from the scan relies on a graph-matching approach whose efficiency proves the relevance of the labeling step, and the flexibility of the assignment approach. Moreover, the use of a graph-matching approach allows the location of a template from its simple design, which is an advantage that can be of real interest for innovating companies that work with innovative components. By improving precision in spatial location, the proposed framework ensures reproducibility during integrated-circuit secure characterization.

To go further, the use of tools proposed in this paper could lead to additional automation that usually requires the human visual system and its understanding abilities. Firstly, the autofocus system could be used for refining the IC positioning on the characterization bench, for example, for tilt adjustment and rotation correction. It could also be useful for focusing the laser beam, visible as a bright spot on the conductive tracks, and for detecting contaminants, which are visible as black spots. Secondly, the scan process that produces a full view of the IC can be computed through different magnifying lenses. These views could be combined to produce a multi-resolution view, as each magnification provides a precision level. The use of such a view could gain considerable time in the characterization of IC.

## Figures and Tables

**Figure 1 sensors-21-02175-f001:**
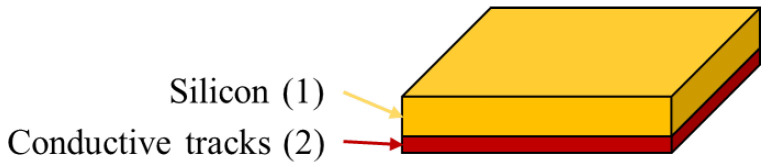
Concept mapping of different materials composing our Integrated Circuit (IC). The conductive tracks (2) are viewed through the silicon layer (1), since the remaining material is opaque to infrared light.

**Figure 2 sensors-21-02175-f002:**
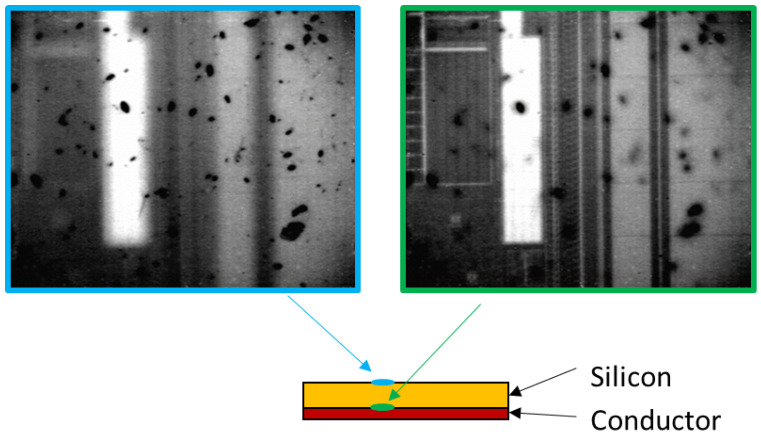
View of contaminants on the silicon surface (blue), shadowing the conductive tracks (green).

**Figure 3 sensors-21-02175-f003:**
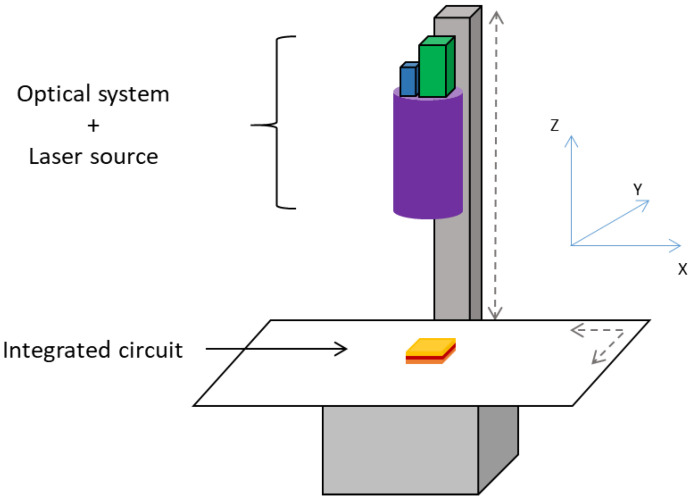
Representation of a bench used for laser-injection-based IC characterization. Each 3D scanning axis is motorized in the 3D XYZ coordinate system. The vertical column represents the *Z* axis, along which the optical system moves thanks to a motor. The horizontal tray, where the IC is placed, is motorized according to the *XY* plane.

**Figure 4 sensors-21-02175-f004:**
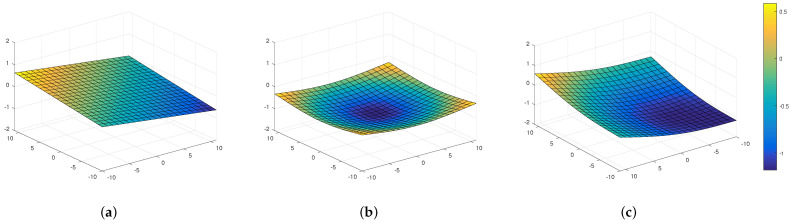
Representation of the surface of a tilted IC (**a**), a curved IC (**b**) and both (**c**).

**Figure 5 sensors-21-02175-f005:**
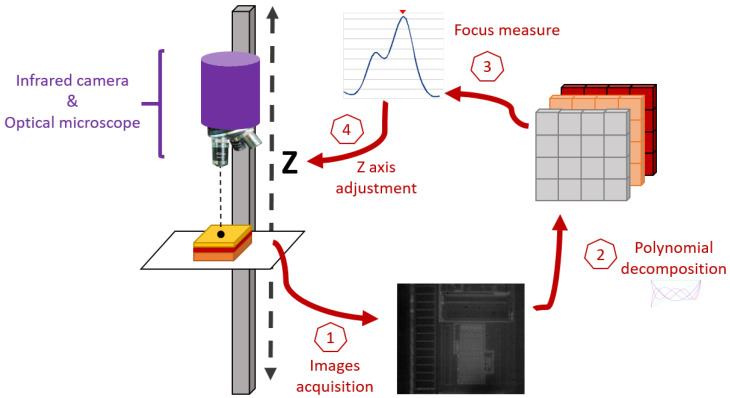
Representation of the autofocus framework using the Focus Metric based on POlynomial Decomposition (FMPOD): images are acquired at different distances to the IC, decomposed to compute the focus measurement that gives the best focus position.

**Figure 6 sensors-21-02175-f006:**
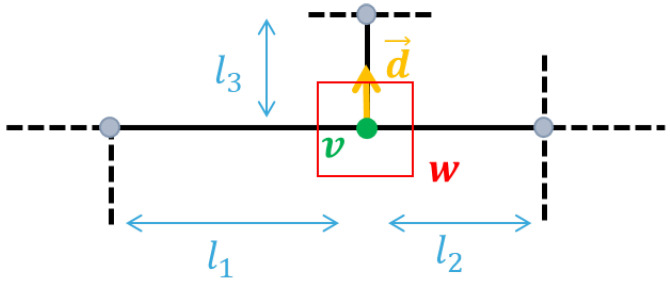
Required elements for invariant computation of our textural descriptor $\alpha_{v}^{H}$ for a node *v*: the histogram is computed for the window *W* around *v*, of size proportional to $min(l_{1},l_{2},l_{3})$, and whose gradients are relative to the main orientation $\vec{d}$ described by its connected edges.

**Figure 7 sensors-21-02175-f007:**
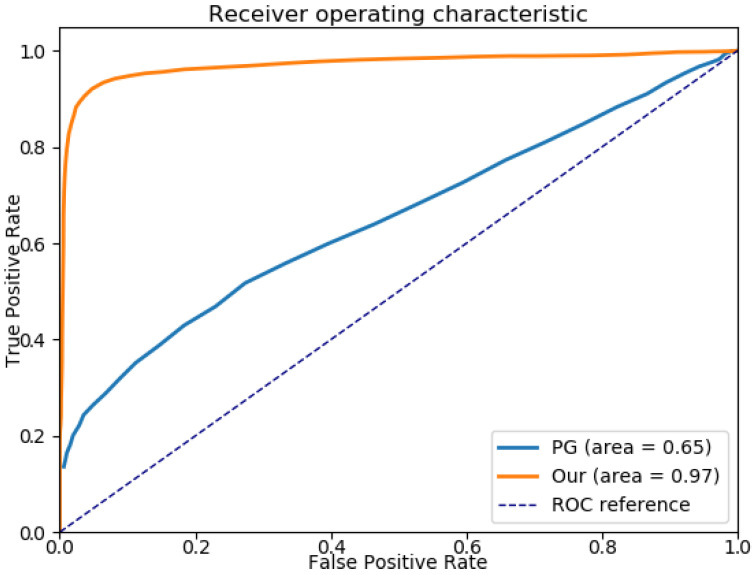
ROC curves for Dutta’s method and our framework. The area under their respective curve represent the overall location success rate.

**Figure 8 sensors-21-02175-f008:**
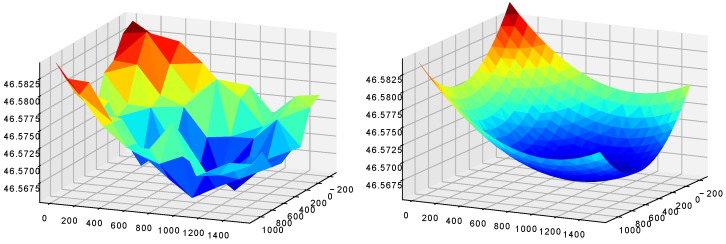
The focus sampling (9 × 9) on the SOI (**left**), and its 3D Gaussian surface approximation (**right**).

**Figure 9 sensors-21-02175-f009:**
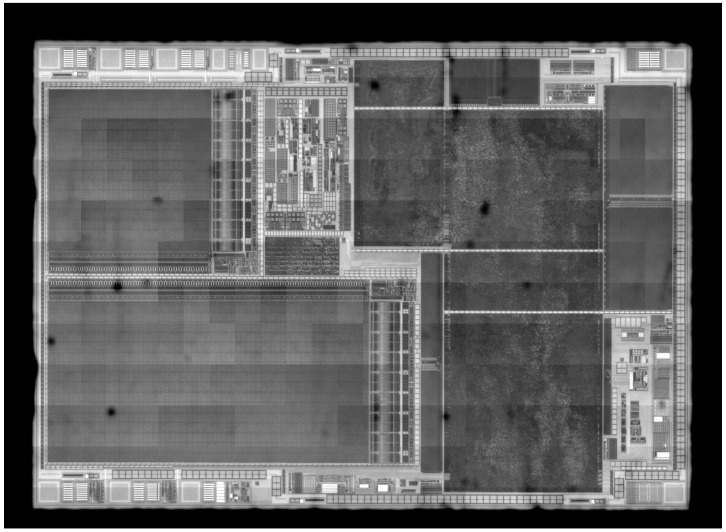
Resulting scan of conductive tracks (the of Interest (SOI) inside an IC under infrared microscope magnification 20×. The black spots correspond to contaminants present on the silicon surface. The small intensity gap between the mosaic images is camera dependent.

**Figure 10 sensors-21-02175-f010:**
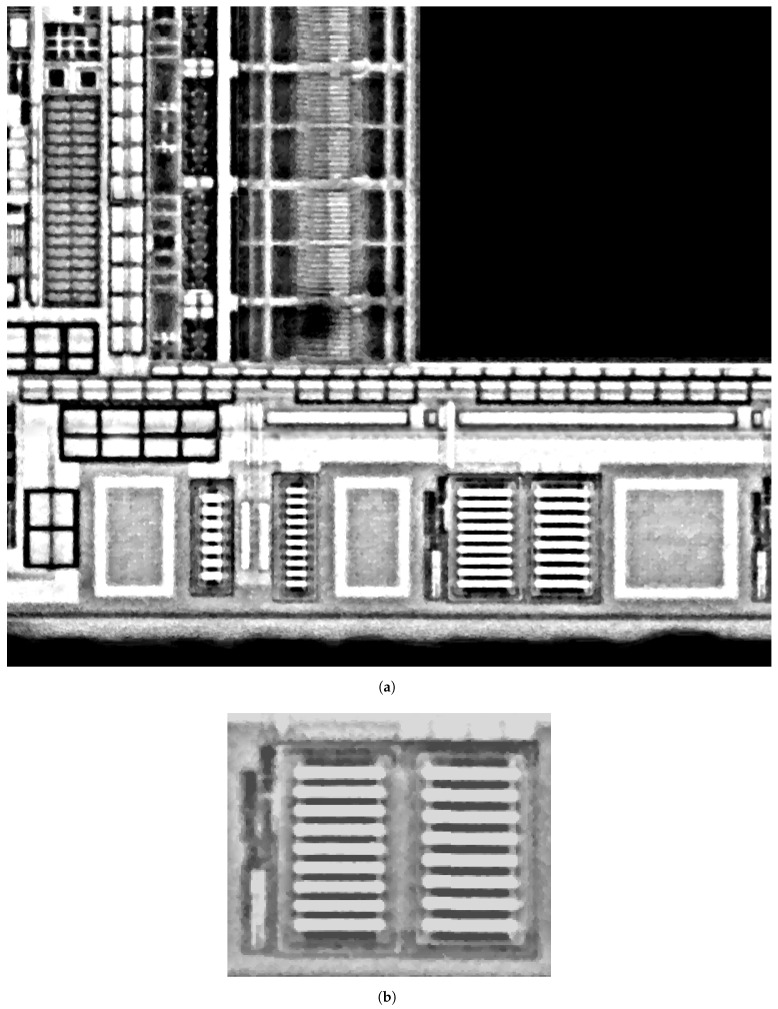
Image acquisition under a 20× optical magnifier of (**a**) a partial view of an integrated circuit (target) (**b**) an electronic structure of interest (template).

**Figure 11 sensors-21-02175-f011:**
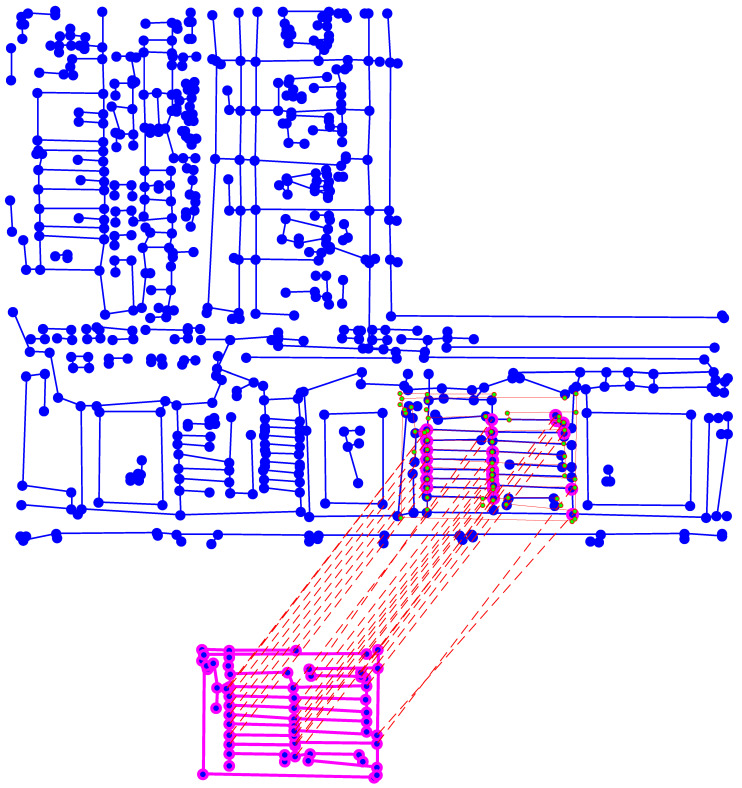
Projection of the template graph $G_{t}$ (**bottom**) located in the target graph $G_{c}$ (**top**). These graphs were created from the images shown in Figure 10. From the matching, only the largest connected sub-graph is kept (highlighted in pink in the target) and used to estimate the template projection. This projection is represented by the graph with green nodes and red edges. $S_{tot}=71\%$.

**Figure 12 sensors-21-02175-f012:**
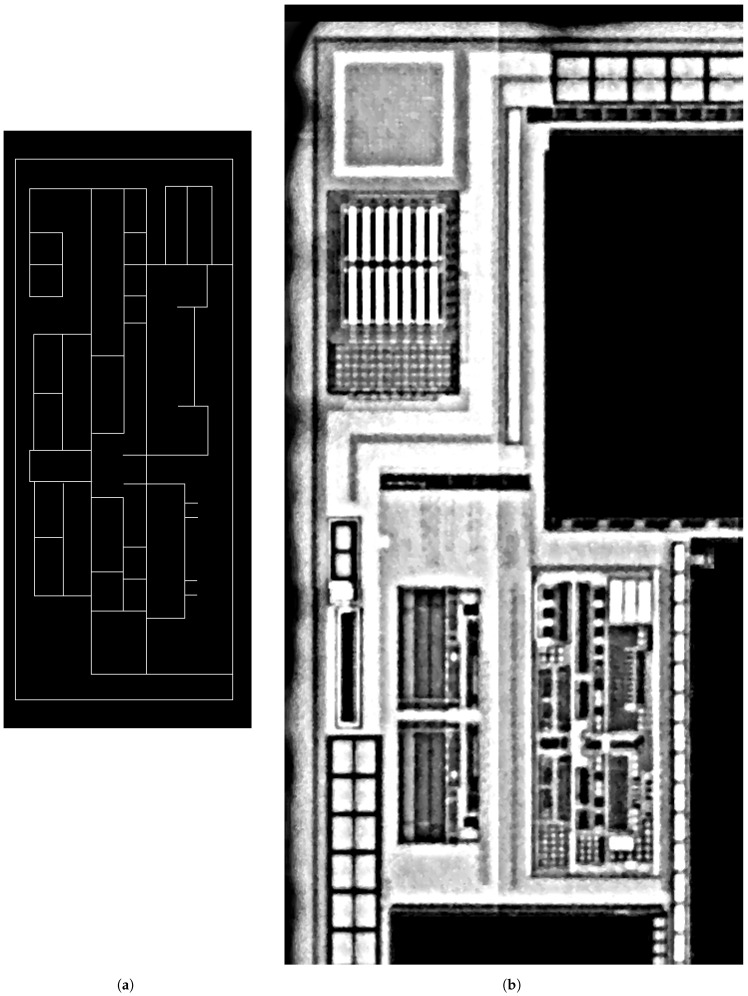
A schematic template (**a**) to match and locate in the target image acquired under a 20× optical magnifier (**b**).

**Figure 13 sensors-21-02175-f013:**
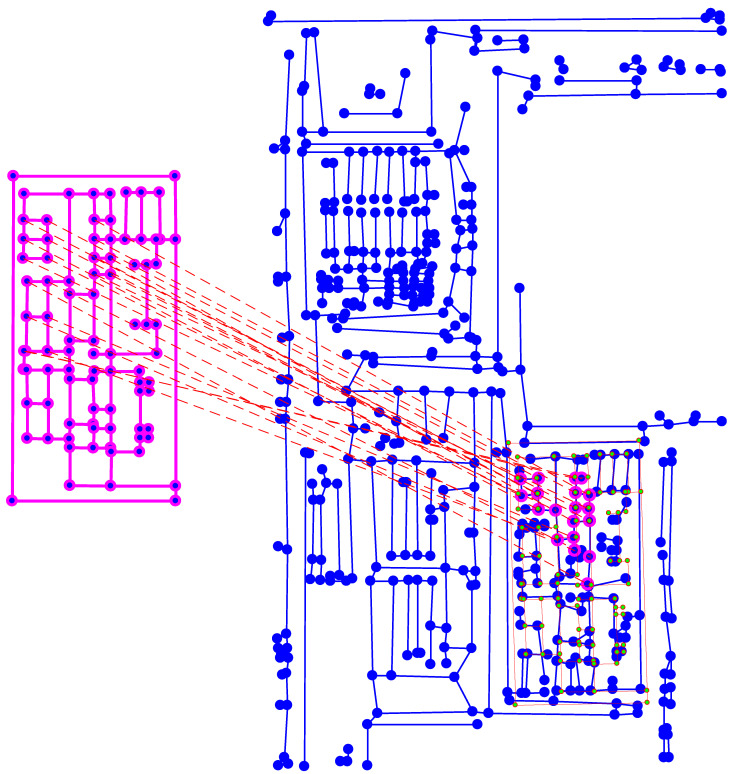
Projection of the template graph (**left**) located in the target graph (**right**). These graphs were created from the images shown in Figure 12. From the matching, only the largest connected sub-graph is kept (highlighted in pink in the target) and used to estimate the template projection. This projection is represented by the graph with green nodes and red edges. $S_{tot}=61\%$.

**Table 1 sensors-21-02175-t001:** Structural descriptor computation for a node *v* and an edge *e* according to their neighborhood in the graph.

	Given a Node *v*	Given an Edge *e*
Let {$E_{i}$} be its # connected edges in decreasing order	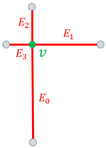	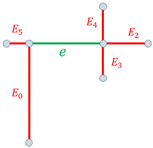
Its structural descriptor	$\alpha_{v}^{S}=\sum_{i=0}^{\#}\left|\right|E_{i}\left|\right|*p_{i}$	$\beta_{e}^{S}=\sum_{i=0}^{\#}\left|\right|E_{i}\left|\right|*p_{i}$

## Data Availability

Data sharing not applicable.

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
