# Peer review of "Spatial Location in Integrated Circuits through Infrared Microscopyâ€"

_sensors, 2021, doi:10.3390/s21062175_

Round 1
Reviewer 1 Report
- {Summary}
A framework is proposed to automate the location in an integrated circuit for its secure characterization. In the framework, a method is proposed to analyze images features through a discrete polynomial image transformation, which followed by a graph matching to locate an interesting structure.
- {Novelty}
Paper contributes some new ideas on the framework of autofocus on IC.
- {Clarity}
Moderate: paper can be improved. The main methods on image graph matching are not well presented, as well as in the Introduction.
- {Evaluation}
Moderate: Experimental results are weak: important baselines are missing, and improvements are not significant, or well presented.
- {Reasons to Reject}
The proposed methods for feature extracting and transformation is not novel. Proposed situation is very similar with many existing methods in image processing area. In the feature extraction and graph matching area, there are various approaches to solve it. The proposed method did not show its advantages over other methods.
- {Detailed Comments}
The proposed framework includes three steps basically: infrared image caption, feature extraction, and matching. As this paper claims, the proposed structural descriptor and textural descriptor works well in the infrared images. However, the graph matching is well studied and applied in many areas, for instance, medical vascular structure graph matching [1] is a similar task, and it takes much simple method. Please give more insight or clarify for the proposed method.
This paper should compare with the existing graph matching methods to judge the effectiveness of the proposed technique.
The paper is not well structured. The first two parts, I. Introduction, II. Scanning system for integrated circuits vision, are the general introduction of the image capture of IC. However, the main work claimed in this paper is the framework, in which a key step is the feature extraction and graph matching. The introduction and key contribution are mixed in the section 3. Patter location, in which the structural descriptor and textural descriptor is the proposed key contribution.
The experiments only show that the proposed methods work in IC images, well, there is no reason to believe that only the proposed method works well, or even better than the existing methods.
[1] Deng, Kexin, et al. "Retinal fundus image registration via vascular structure graph matching." International Journal of Biomedical Imaging 2010 (2010).
Reviewer 2 Report
Authors proposed new approach for structure location of the integrated circuits (ICs) using infrared microscopy. As far as I know, industry has various approaches to find faults and cracks because of the cost issues in the fabrication and testing process. This is a kind of interesting approaches and it could be useful for the industry IC companies.
Literature review is good to understand previous scanning systems for ICs. There are no English grammar mistakes. Authors provided procedures to use how the scanning process can be used for structure locations of the ICs. Authors used proper X,Y,Z, and angle movement algorithms for scanning data using microscopy. Authors also showed the ROC curve which is good index for the measurement data of the scanning process.
However, in my only critical concerns, there are no references in the introduction sections in 1st, 2nd, and 3rd page. As far as I know, this is industry knowledge so it is hard to find the journal or conference papers. However, authors can find some references from US(United states) and JP(Japan) patents in the google scholar.
Therefore, the manuscript could be minor revision. There are some minor comments as below.
- Authors need to provide the detail information for the conference such as city, country, and date, and page numbers.
- Authors need to use proper abbreviated journal name.
- Please increase quality of Figure 3.
- Please use proper Table format.
- Please use unit of Figure 8.
- Please use same fonts In the list of acronyms.
- Please do not use italics in Funding information.
Reviewer 3 Report
The authors presented an interesting approach for the inspection of integrated circuits by infrared microscopy mounted on a 3-axes scanning system. The main challenges were to properly focus the optics and to develop an effective pattern recognition method.
My main concern is in relation to English language. Although understandable, the text has several grammar errors and some sentences need to be rephrased. I am including a pdf copy with some comments/suggestions for your consideration (up to page 8). I recommend to thoroughly revise English grammar and re-submit.

Author Response
The authors presented an interesting approach for the inspection of integrated circuits by infrared microscopy mounted on a 3-axes scanning system. The main challenges were to properly focus the optics and to develop an effective pattern recognition method.
My main concern is in relation to English language. Although understandable, the text has several grammar errors and some sentences need to be rephrased. I am including a pdf copy with some comments/suggestions for your consideration (up to page 8). I recommend to thoroughly revise English grammar and re-submit.
Response :
The English grammar has been thoroughly reworked. Please find these corrections highlighted in the revision.
Thank you for your review.
Round 2
Reviewer 3 Report
The authors provided a corrected version of their work that I find adequate for publication.